# Superimposed Perfect Binary Array-Aided Channel Estimation for OTFS Systems

**DOI:** 10.3390/e25081163

**Published:** 2023-08-03

**Authors:** Zuping Tang, Hengyou Kong, Ziyu Wu, Jiaolong Wei

**Affiliations:** School of Electronic Information and Communications, Huazhong University of Science and Technology, Wuhan 430074, China; tang_zuping@hust.edu.cn (Z.T.); m202172411@hust.edu.cn (H.K.); wzychd@163.com (Z.W.)

**Keywords:** orthogonal time-frequency space modulation, channel estimation, superimposed pilot, perfect binary array

## Abstract

Orthogonal time-frequency space (OTFS) modulation outperforms orthogonal frequency-division multiplexing in high-mobility scenarios through better channel estimation. Current superimposed pilot (SP)-based channel estimation improves the spectral efficiency (SE) when compared to that of the traditional embedded pilot (EP) method. However, it requires an additional non-superimposed EP delay-Doppler frame to estimate the delay-Doppler taps for the following SP-aided frames. To handle this problem, we propose a channel estimation method with high SE, which superimposes the perfect binary array (PBA) on data symbols as the pilot. Utilizing the perfect autocorrelation of PBA, channel estimation is performed based on a linear search to find the correlation peaks, which include both delay-Doppler tap information and complex channel gain in the same superimposed PBA frame. Furthermore, the optimal power ratio of the PBA is then derived by maximizing the signal-to-interference-plus-noise ratio (SINR) to optimize the SE of the proposed system. The simulation results demonstrate that the proposed method can achieve a similar channel estimation performance to the existing EP method while significantly improving the SE.

## 1. Introduction

As one of the most widely adopted multicarrier modulation systems, orthogonal frequency-division multiplexing (OFDM) offers the advantages of anti-frequency selectivity and high spectral efficiency (SE). In high-mobility scenarios, however, time-varying channels with high Doppler spread severely affect OFDM performance [1,2,3].

Hence, in [4,5], R. Hadani et al. derived orthogonal time-frequency space (OTFS) modulation, which can address the above limitations. Unlike classic OFDM, by mapping data symbols into the delay-Doppler (DD) domain instead of the time-frequency (TF) domain [6], OTFS converts doubly selective channels into stable and sparse channels in the DD domain. In this way, all transmitted symbols experience near-constant channel gain [7,8]. Therefore, at the receiver side, the complexity of channel estimation and data detection is reduced. Furthermore, OTFS is compatible with existing multicarrier modulations through pre- and post-processing modules [9,10,11].

Accurate channel estimation at the receiver is a precondition for OTFS data detection. The current channel estimation methods can be roughly divided into three categories. The first category uses the entire OTFS frame as the pilot for channel estimation, and the estimated information is used for data detection in next frame, which can be represented by the method of [12]. However, the performance of these methods may be degraded because the channel estimation easily becomes outdated in the following frame for rapidly time-varying channels.

The second category is the embedded pilot (EP) channel estimation method proposed in [13]. By embedding a single pilot symbol in the DD domain and setting an appropriate threshold for its response, high-precision channel estimation is achieved. In [14,15,16], the EP concept was redesigned to improve the channel estimation performance. However, guard symbols are always needed in this channel estimation category to avoid interference between pilot and data symbols, which results in SE loss.

The third category is the superimposed pilot (SP)-based channel estimation method proposed in [17,18]. In this method, the pilot symbols are superimposed onto the data symbols, and the guard symbols in the second category are no longer needed. This mitigates the SE loss. In the method of [17], the complex channel gains were estimated by treating the data as interference; in addition, iteration between channel estimation and data detection was adopted to prevent performance degradation at high SNRs. However, the delay-Doppler taps of the channel must be estimated by means of an additional non-superimposed EP frame for the following SP-aided frames, as they can only estimate the complex channel gains in this method. In rapidly delay-Doppler varying communication scenarios, this SP method exhibits lower adaptability when compared to that of the EP method. As an alternative approach, Ref. [18] proposed a method that utilized the whole frame for data transmission, and a single pilot symbol was superimposed onto the data symbol. This pilot design is similar to the EP method in the second category. In [18], no dedicated DD domain grids were occupied for channel estimation; however, since pilot and data symbols were superimposed, the interference between them was unavoidable. As a result, an iterative process must be adopted to mitigate this interference.

In this paper, we propose an OTFS channel estimation method based on the superimposed perfect binary array (SPBA). The main contributions of this paper are as follows:(i)In the proposed SPBA method, the whole OTFS frame is used for data transmission, while a perfect binary array (PBA) that plays the role of the pilot is superimposed on data symbols in the DD domain, resulting in higher SE when compared to that of the EP scheme in [11]. Furthermore, in contrast to similar work in [17], the SPBA method can estimate delay-Doppler taps and complex path gains in the same superimposed frame, which means the additional non-superimposed EP frame is no longer required. As a result, the SPBA method has better adaptability to rapidly delay-Doppler varying channels.(ii)A channel estimator for the SPBA framework is proposed. At the receiver side, local PBAs with different circular shifts in the DD domain are correlated with the received signal. By utilizing the perfect autocorrelation of PBA [19,20,21], we can find the propagation paths and determine the delay-Doppler taps together with the complex path gains by comparing the correlation values to a threshold. Hence, channel estimation is performed through a linear search for the correlation peak. The proposed channel estimator can achieve high estimation accuracy with a significant SE improvement.(iii)The SPBA’s optimal power ratio is calculated by deriving the signal-to-interference-plus-noise ratio (SINR) and maximizing it. We validate that the adoption of the optimal power ratio will further improve SE and bit error rate (BER) performance of the SPBA method.

We validate the effectiveness of the proposed method and show that this method can approach the channel estimation performance of the EP method while significantly improving the SE.

The remainder of this paper is organized as follows: Section 2 formalizes the superimposed PBA-aided OTFS (SPBA-OTFS) system model and derives the corresponding channel input–output relation. Section 3 proposes the PBA-based channel estimation algorithm and analyzes its modifications in the case of OTFS modulation using rectangular waveforms. Section 4 analyzes the SE of the proposed system and derives the optimal power ratio by means of maximizing the SINR. Simulation results are presented in Section 5, followed by conclusions in Section 6.

## 2. System Model

We consider a set of MN modulated data symbols $\left\{x_{d}\left[k,l\right],k\in \left[0,N-1\right],l\in \left[0,M-1\right]\right\}$ placed in DD-domain grids, where *k* and *l* represent Doppler shift and delay indices, respectively. The PBA symbol $x_{p}\left[k,l\right]$, which will be introduced later, is superimposed on $x_{d}\left[k,l\right]$ to yield the superimposed symbol x[k,l] in the DD domain for transmission, as follows:(1)$$x\left[k,l\right]=x_{d}\left[k,l\right]+x_{p}\left[k,l\right],\;0\leq k\leq N-1,\;0\leq l\leq M-1.$$

The MN PBA symbols in a single frame form a two-dimensional (2D) array P that plays the role of pilot. The power levels of PBA and data symbols are assumed to be $\rho_{p}=E\left\{{\left|x_{p}\left[k,l\right]\right|}^{2}\right\}=\beta \rho$ and $\rho_{d}=E\left\{{\left|x_{d}\left[k,l\right]\right|}^{2}\right\}=\left(1-\beta \right)\rho$, respectively, where the zero-mean data symbols $x_{d}\left[k,l\right]$ are independent and identically distributed (i.i.d.), ρ is the power of each superimposed symbol, and β∈[0,1] denotes the power ratio between PBA and superimposed symbols.

According to (Equation 1), the frame structure in SPBA-OTFS systems is illustrated in Figure 1b. The proposed method enables the whole OTFS frame to be used for data transmission when compared to the classic EP scheme from [13] shown in Figure 1a. The EP scheme embeds the single pilot symbol at $\left(k_{p},l_{p}\right)$ and arranges the guard symbols between data and pilot symbols, i.e.,
(2)$$x\left[k,l\right]=\left\{\begin{array}{cc} x_{p}\left[k,l\right] & k=k_{p},l=l_{p},\\ 0 & k_{p}-2k_{max}\leq k\leq k_{p}+2k_{max},\\ & l_{p}-l_{max}\leq l\leq l_{p}+l_{max},\\ x_{d}\left[k,l\right] & \;\mathrm{otherwise}.\; \end{array}\right.$$
where $k_{max}$ and $l_{max}$ denote the taps corresponding to the maximum Doppler and delay values, respectively. In the proposed method, the guard symbols are no longer needed as PBA symbols are superimposed on data symbols. Therefore, the SE is radically improved.

Furthermore, Figure 2 presents a comparison of the transmitted data frames between the SP method from [17] and the proposed SPBA method. The SP method shown in Figure 2a requires an additional non-superimposed EP frame in Figure 1a to estimate the delay-Doppler taps for the following SP frames. In contrast, the proposed SPBA method can avoid the use of this dedicated pilot frame by utilizing the superimposed PBA to estimate delay-Doppler taps and complex path gains in the same SPBA frame, as shown in Figure 2b.

Then, the MN superimposed symbols in the DD domain are mapped to the TF domain through an inverse symplectic finite Fourier transform (ISFFT), expressed as
(3)$$X\left[n,m\right]=\frac{1}{\sqrt{NM}}\sum_{k=0}^{N-1}\sum_{l=0}^{M-1}x\left[k,l\right]e^{j2\pi \left(\frac{nk}{N}-\frac{ml}{M}\right)},$$
where n∈[0,N−1] and m∈[0,M−1] are time and subcarrier indices, respectively. Next, the values of X[n,m] are converted into a continuous-time waveform s(t) via Heisenberg transformation, i.e.,
(4)$$s\left(t\right)=\sum_{n=0}^{N-1}\sum_{m=0}^{M-1}X\left[n,m\right]g_{\mathrm{tx}}\left(t-nT\right)e^{j2\pi m\Delta f(t-nT)},$$
where *T* and Δf denote the symbol duration and subcarrier spacing, respectively, and $g_{\mathrm{tx}}\left(t\right)$ is the transmit pulse.

Doubly selective channels are stable and sparse in the DD domain, and the channel impulse response is given by
(5)$$h\left(\tau ,v\right)=\sum_{i=1}^{P}h_{i}\delta \left(\tau -\tau_{i}\right)\delta \left(v-v_{i}\right),$$
where *P* is the number of propagation paths and $h_{i}$ is the complex channel gain of the *i*-th path, distributed as $CN\left(0,\sigma_{h_{i}}^{2}\right)$ [13], which denotes the complex Gaussian random variable with zero mean and variance $\sigma_{h_{i}}^{2}$. $\tau_{i}$ and $v_{i}$ denote the delay and Doppler values, respectively, of the *i*-th path. They can be expressed in terms of the delay index $l_{i}$ and Doppler index $k_{i}$ as $\tau_{i}=l_{i}/M\Delta f$ and $v_{i}=k_{i}/NT$. In this paper, only integer Doppler values are considered. For the fractional Doppler scenario, the proposed method is still valid if the correlation interval on the Doppler axis is increased.

Considering the effects of the multipath time-varying channel conditions, the received signal can be expressed as
(6)$$r\left(t\right)=\;\int \int \;h\left(\tau ,v\right)e^{j2\pi v(t-\tau )}s\left(t-\tau \right)d\tau dv+n\left(t\right),$$
where n(t) is additive Gaussian white noise distributed as $CN\left(0,\sigma_{n}^{2}\right)$.

At the receiver side, the received symbols Y[n,m] in the TF domain are obtained by applying the Wigner transform to r(t) with the received pulse $g_{\mathrm{rx}}\left(t\right)$. Finally, by performing the symplectic finite Fourier transform (SFFT) on Y[n,m], the received symbols are converted back to the DD domain, where they are expressed as y[k,l]. According to [11], the relationship between the transmitted and received symbols in the DD domain is
(7)$$y\left[k,l\right]=\sum_{i=1}^{P}h_{i}e^{-j2\pi \frac{l_{i}k_{i}}{MN}}\alpha_{i}\left[k,l\right]x\left[{\left(k-k_{i}\right)}_{N},{\left(l-l_{i}\right)}_{M}\right]+\omega \left[k,l\right],$$
where ω[k,l] denotes the noise in the DD domain, distributed as $CN\left(0,\sigma_{\omega }^{2}\right)$, and the subscript ${\left(\cdot \right)}_{N}$ denotes the modulo operation with divisor *N*. $\alpha_{i}\left[k,l\right]=1$ when the transmitted and received pulses are ideal, meaning that they satisfy the biorthogonality conditions [11]. Otherwise, affected by the imperfect biorthogonality of the rectangular pulses, $\alpha_{i}\left[k,l\right]$ is expressed as
(8)$$\alpha_{i}\left[k,l\right]=\left\{\begin{array}{c} e^{j2\pi \frac{lk_{i}}{MN}},\;l_{i}\leq l<M,\\ \frac{N-1}{N}e^{j2\pi \left(\frac{lk_{i}}{MN}-\frac{{\left(k-k_{i}\right)}_{N}}{N}\right)},\;0<l<l_{i}. \end{array}\right.$$

By substituting (Equation 1) into (Equation 7) and re-expressing the equation in matrix form, the received symbol matrix of an SPBA-OTFS system is given by
(9)$$Y=\sum_{i=1}^{P}h_{i}e^{-j2\pi \frac{l_{i}k_{i}}{MN}}A_{i}\circ \left(D_{k_{i},l_{i}}+P_{k_{i},l_{i}}\right)+W,$$
where $Y\in C^{N\times M}$, $W\in C^{N\times M}$, $A_{i}\in C^{N\times M}$, $D_{k_{i},l_{i}}\in C^{N\times M}$ and $P_{k_{i},l_{i}}\in C^{N\times M}$ are the matrix forms of y[k,l], ω[k,l], $\alpha_{i}\left[k,l\right]$, $x_{d}\left[{\left(k-k_{i}\right)}_{N},{\left(l-l_{i}\right)}_{M}\right]$ and $x_{p}\left[{\left(k-k_{i}\right)}_{N},{\left(l-l_{i}\right)}_{M}\right]$, respectively. For two matrices A and B, the operation A∘B denotes their Hadamard product.

## 3. Superimposed Perfect Binary Array-Aided OTFS Channel Estimator

### 3.1. Proposed Channel Estimator

According to (Equation 9), the received symbols can be considered equivalent to weighted superpositions of the transmitted symbols with different circular shifts. Hence, in an SPBA-OTFS system, the PBA symbols also have the same shifts, which are determined by the delay and Doppler indices of the paths. As a result, the estimation of the delay and Doppler shifts can be transformed into a linear search of the indices. Therefore, a 2D linear search is performed in the DD domain at the receiver side by correlating the received symbol matrix Y with the local PBA Γ over all possible delay and Doppler indices.

Furthermore, the step sizes for the linear search of the delay and Doppler indices are set as the quantization steps of the delay axis (1/MΔf) and the Doppler axis (1/NT), respectively. For each search operation, we define a search unit *J*, which is associated with delay value ($l_{J}/M\Delta f$) and Doppler value ($k_{J}/NT$), where $l_{J}\in \left[0,l_{m}\right]$ and $k_{J}\in \left[-k_{m},k_{m}\right)$ denote the delay and Doppler indices of the local PBA, respectively. Here, $l_{m}$ and $k_{m}$ are the maximum delay and Doppler shift indices of the channel, respectively.

In this paper, we first assume an SPBA-OTFS system with ideal pulses, meaning that $\alpha_{i}\left[k,l\right]=1$ in (Equation 7). In this case, the superimposed PBA affected by the *q*-th path can be expressed as
(10)$$\Lambda_{q}=h_{q}e^{-j2\pi \frac{l_{q}k_{q}}{MN}}P_{k_{q},l_{q}},$$
where 1≤q≤P. Following (Equation 10), we define the local PBA associated with search unit *J* as
(11)$$\Gamma_{J}=\frac{e^{j2\pi \frac{l_{J}k_{J}}{MN}}P_{k_{J},l_{J}}^{*}}{\beta \rho },$$
where $P_{k_{J},l_{J}}^{*}$ denotes the conjugate of the matrix $P_{k_{J},l_{J}}^{}$.

For channel estimation, we correlate Y with $\Gamma_{J}$ at the receiver side. The correlation value $z\left[k_{J},l_{J}\right]$ can be expressed as
(12)$$z\left[k_{J},l_{J}\right]=\frac{\mathrm{sum}\left(Y\circ \Gamma_{J}\right)}{MN},$$
where the operation $\mathrm{sum}(Y\circ \Gamma_{J})$ computes the sum of all elements of the matrix $Y\circ \Gamma_{J}$.

When the delay and Doppler indices associated with the search unit *J* are equal to those of the *q*-th path, which means that $\left(k_{J},l_{J}\right)=\left(k_{q},l_{q}\right)$, $\Gamma_{J}$ has the maximum correlation with $\Lambda_{q}$. Consequently, a peak that reflects the channel state information (CSI) appears after the correlation calculation. In this condition, the correlation value is
(13)$$z\left[k_{J},l_{J}\right]=z\left[k_{q},l_{q}\right]=h_{q}+v_{q},$$
where $h_{q}$ is the complex gain of the *q*-th path and $v_{q}$ denotes the interference term in the correlation value, given by
(14)$$v_{q}=v_{q,p}+v_{q,d}+v_{q,\omega },$$
where $v_{q,p}$, $v_{q,d}$ and $v_{q,\omega }$ denote the interference between the PBAs affected by different channels, between PBA and the data symbol matrix, and between PBA and the noise matrix, respectively. They can be expressed as
(15)$$v_{q,p}=\frac{\sum_{i=1,i\neq q}^{P}h_{i}e^{j2\pi \frac{\left(l_{q}-l_{i}\right)\left(k_{q}-k_{i}\right)}{MN}}\mathrm{sum}\left(P_{k_{i},l_{i}}\circ P_{k_{q},l_{q}}^{*}\right)}{MN(\beta \rho )},$$
(16)$$v_{q,d}=\frac{\sum_{i=1}^{P}h_{i}e^{j2\pi \frac{\left(l_{q}-l_{i}\right)\left(k_{q}-k_{i}\right)}{MN}}\mathrm{sum}\left(D_{k_{i},l_{i}}\circ P_{k_{q},l_{q}}^{*}\right)}{MN(\beta \rho )},$$
(17)$$v_{q,\omega }=\frac{e^{j2\pi \frac{l_{q}k_{q}}{MN}}\mathrm{sum}\left(W\circ P_{k_{q},l_{q}}^{*}\right)}{MN(\beta \rho )}.$$

Furthermore, considering the autocorrelation of PBA, which will be introduced later in this section, we have
(18)$$\mathrm{sum}\left(P_{k_{i},l_{i}}\circ P_{k_{j},l_{j}}^{*}\right)=\left\{\begin{array}{c} 0\left(i=j\right)\\ MN\left(i\neq j\right) \end{array}\right.,$$
as a result, $v_{q,p}=0$ in (Equation 15), indicating that there is no interference between the PBAs affected by different channels according to the perfect autocorrelation of PBA.

When delay and Doppler indices associated with search unit *J* are not equal to those of any propagation path, $\Gamma_{J}$ has no correlation with the received symbol matrix Y. In this condition, only the interference terms are included in $z\left[k_{J},l_{J}\right]$. According to (Equation 12) and (Equation 18), in this condition, $z\left[k_{J},l_{J}\right]$ is given as
(19)$$z\left[k_{J},l_{J}\right]=v_{J,d}+v_{J,\omega }.$$

In summary, the correlation value can be expressed as
(20)$$z\left[k_{J},l_{J}\right]=\left\{\begin{array}{cc} h_{J}+v_{J,d}+v_{J,\omega } & \mathrm{if}\;\left(k_{J},l_{J}\right)\;\text{is the delay-Doppler indices of the}\;J\text{-th path}\\ v_{J,d}+v_{J,\omega } & \mathrm{else} \end{array}\right.$$

The perfect autocorrelation of PBA leads to a significant amplitude difference in $\left|z\left[k_{J},l_{J}\right]\right|$ between the two conditions. Therefore, $\left|z\left[k_{J},l_{J}\right]\right|$ is compared to a threshold γ; if $\left|z\left[k_{J},l_{J}\right]\right|>\gamma$, we consider this search unit *J* to be associated with a valid path in the channel. Supposing that it is the *q*-th path, we have $\left(k_{J},l_{J}\right)=\left(k_{q},l_{q}\right)$, and the path gain estimate is
(21)$${\hat{h}}_{q}=z\left[k_{J},l_{J}\right]=h_{q}+v_{q,d}+v_{q,\omega }.$$

The threshold γ corresponds to the mean squared error (MSE) of the path gain estimate. The MSE of ${\hat{h}}_{q}$ is given as
(22)$$\begin{array}{cc} \mathrm{MSE}({\hat{h}}_{q}) & =\sigma_{v_{q,d}+v_{q,\omega }}^{2}\\ \; & =E\left\{\left(v_{q,d}+v_{q,\omega }\right)\right.\left.\cdot {\left(v_{q,d}+v_{q,\omega }\right)}^{*}\right\}\\ \; & =\frac{\left(\frac{\sigma_{H}^{2}\left(1-\beta \right)}{\beta }+\frac{\sigma_{\omega }^{2}}{\beta \rho }\right)}{MN}, \end{array}$$
where $\sigma_{H}^{2}=\sum_{q=1}^{P}\sigma_{h_{q}}^{2}$. According to (Equation 22), the MSE of each path gain can be rewritten as $\mathrm{MSE}(\hat{h})$ because it is independent of the specific path. As a result, the MSE of the threshold-based channel estimation method is given as
(23)$$\sigma_{\mathrm{Thr}}^{2}=\sum_{q=1}^{P}\sigma_{v_{q}}^{2}=P\cdot \mathrm{MSE}\left(\hat{h}\right).$$

The threshold satisfies $\gamma^{2}\propto \mathrm{MSE}\left(\hat{h}\right)$. Algorithm 1 summarizes the proposed SPBA-OTFS channel estimation method.
**Algorithm 1** Superimposed Perfect Binary Array-Aided OTFS Channel Estimation Method**Input:** Received symbol matrix Y, PBA Γ, maximum delay $l_{m}$, maximum Doppler shift $k_{m}$, threshold γ
**Output:** Channel estimates $\left\{\left({\hat{h}}_{1},k_{1},l_{1}\right),\cdots ,\left({\hat{h}}_{i},k_{i},l_{i}\right)\right\}$
  1:  **Initialize:** path number i=1
  2:  **for** $k_{J}=-k_{m}$ to $k_{m}-1$ **do**
  3:    **for** $l_{J}=0$ to $l_{m}$ **do**
  4:    Generate the local PBA $\Gamma_{k_{J},l_{J}}$ via (Equation 11);
  5:    Compute the correlation value $z\left[k_{J},l_{J}\right]$ via (Equation 12);
  6:    **if** $\left|z\left[k_{J},l_{J}\right]\right|>\gamma$ **then**
  7:     Update i=i+1;
  8:     ${\hat{h}}_{i}\leftarrow \left|z\left[k_{J},l_{J}\right]\right|$, $k_{i}\leftarrow k_{J}$, $l_{i}\leftarrow l_{J}$;
  9:    **end if**
10:   **end for**
11:  **end for**


### 3.2. Design of the PBA

According to Section 3.1, we wish to find a P that satisfies (Equation 18) to expand the amplitude difference in $\left|z\left[k_{J},l_{J}\right]\right|$ between the conditions where channels are successfully estimated or not.

According to [19], a 2D PBA will meet this requirement. A matrix $A\in C^{N\times M}$ is called a PBA if it satisfies the following conditions: (i) each element of A is ±1, and (ii) the periodic autocorrelation function (PACF) of A is 0, which means that
(24)$$\begin{array}{cc} \mathrm{PACF}(u,v) & =\sum_{i=0}^{N-1}\sum_{j=0}^{M-1}a\left[i,j\right]\cdot a\left[{\left(i-u\right)}_{N},{\left(j-v\right)}_{M}\right]\\ \; & =\mathrm{sum}(A\circ A_{u,v})\\ \; & =\left\{\begin{array}{c} NM,\begin{array}{c} \left(u,v)=(0,0\right) \end{array}\\ 0,\;\;\;\begin{array}{c} \begin{array}{c} \left(u,v)\neq (0,0\right) \end{array} \end{array} \end{array}\right., \end{array}$$
where a[i,j] denotes the (i,j)-th element of A.

According to (Equation 24), if the superimposed array P is a PBA, then (Equation 18) can be easily satisfied. Next, we will clarify how the PBA is created.

Two kinds of 2D binary arrays, defined as follows, are used for the iterative construction of the PBA [20].

A matrix $B\in C^{N\times M}$ is called a quasiperfect binary array (QPBA) if it satisfies the following conditions: (i) each element of B is ±1, and (ii) the periodic quasi-autocorrelation function (QACF) of B is 0, which means that
(25)$$\begin{array}{cc} \mathrm{QACF}(u,v) & =\sum_{i=0}^{N-u-1}\sum_{j=0}^{M-1}b\left[i,j\right]\cdot b\left[{\left(i-u\right)}_{N},{\left(j-v\right)}_{M}\right]\\ \; & -\sum_{i=N-u}^{N-1}\sum_{j=0}^{M-1}b\left[i,j\right]\cdot b\left[{\left(i-u\right)}_{N},{\left(j-v\right)}_{M}\right]\\ \; & =\left\{\begin{array}{c} NM,\begin{array}{c} \left(u,v)=(0,0\right) \end{array}\\ 0,\;\;\;\begin{array}{c} \begin{array}{c} \left(u,v)\neq (0,0\right) \end{array} \end{array} \end{array}\right., \end{array}$$
where b[i,j] denotes the (i,j)-th element of B.

A matrix $C\in C^{N\times M}$ is called a doubly quasiperfect binary array (DQPBA) if it satisfies the following conditions: (i) each element of C is ±1, and (ii) the periodic doubly quasi-autocorrelation function (DQACF) of C is 0, which means that
(26)$$\begin{array}{cc} \mathrm{DQACF}(u,v) & =\sum_{i=0}^{N-u-1}\sum_{j=0}^{M-vs.-1}c\left[i,j\right]\cdot c\left[{\left(i-u\right)}_{N},{\left(j-v\right)}_{M}\right]\\ \; & -\sum_{i=N-u}^{N-1}\sum_{j=0}^{M-vs.-1}c\left[i,j\right]\cdot c\left[{\left(i-u\right)}_{N},{\left(j-v\right)}_{M}\right]\\ \; & -\sum_{i=0}^{N-u-1}\sum_{j=M-v}^{M-1}c\left[i,j\right]\cdot c\left[{\left(i-u\right)}_{N},{\left(j-v\right)}_{M}\right]\\ \; & -\sum_{i=N-u}^{N-1}\sum_{j=M-v}^{M-1}c\left[i,j\right]\cdot c\left[{\left(i-u\right)}_{N},{\left(j-v\right)}_{M}\right]\\ \; & =\left\{\begin{array}{c} NM,\begin{array}{c} \left(u,v)=(0,0\right) \end{array}\\ 0,\;\;\;\begin{array}{c} \begin{array}{c} \left(u,v)\neq (0,0\right) \end{array} \end{array} \end{array}\right., \end{array}$$
where c[i,j] denotes the (i,j)-th element of C.

There are four common ways to iteratively construct a high-order PBA, QPBA, and DQPBA using their low-order forms [21].

(i)Suppose that $A\in C^{N\times M}$ and $B\in C^{N\times M}$ are PBA and QPBA, respectively. A new PBA $A^{\prime }\in C^{2N\times 2M}$ can be constructed as follows:
(27)$$a^{\prime }\left[i,j\right]=\left\{\begin{array}{c} \;a\left[{\left(i\right)}_{N},j/2\right],{\;\;\;\;\;\;(j)}_{2}\equiv 0\\ b\left[{\left(i\right)}_{N},\left(j-1\right)/2\right],{\;\;(j)}_{2}\equiv 1,i<N\\ -b\left[{\left(i\right)}_{N},\left(j-1\right)/2\right],{\left(j\right)}_{2}\equiv 1,i\geq N \end{array}\right.,$$
where $a^{\prime }\left[i,j\right]$ denotes the (i,j)-th element of $A^{\prime }$, i∈[0,2N−1] and j∈[0,2M−1].(ii)Suppose that $B\in C^{N\times M}$ and $C\in C^{N\times M}$ are QPBA and DQPBA, respectively. A new QPBA $B^{\prime }\in C^{2N\times 2M}$ can be constructed as follows:
(28)$$b^{\prime }\left[i,j\right]=\left\{\begin{array}{c} \;b\left[i/2,{\left(j\right)}_{M}\right],{\;\;\;\;\;(i)}_{2}\equiv 0\\ c\left[\left(i-1\right)/2,{\left(j\right)}_{M}\right],{\;\;(i)}_{2}\equiv 1,j<M\\ -b\left[\left(i-1\right)/2,{\left(j\right)}_{M}\right],{\left(i\right)}_{2}\equiv 1,j\geq M \end{array}\right.,$$
where $b^{\prime }\left[i,j\right]$ denotes the (i,j)-th element of $B^{\prime }$, i∈[0,2N−1] and j∈[0,2M−1].(iii)Suppose that $B\in C^{N\times M}$ is a QPBA and that $\left(M/\gcd(N,M)\right)\;\mathrm{mod}\;2\equiv 1$, where the operator gcd(N,M) outputs the greatest common divisor of *N* and *M*. A new DQPBA $C^{\prime }\in C^{N\times M}$ can be constructed as follows:
(29)$$c^{\prime }\left[i,j\right]=\left\{\begin{array}{c} \;b\left[{\left(i-j\right)}_{N},j\right],\left(i-j\right)\geq 0\\ -b\left[{\left(i-j\right)}_{N},j\right],\left(i-j\right)<0 \end{array}\right.,$$
where $c^{\prime }\left[i,j\right]$ denotes the (i,j)-th element of $C^{\prime }$, i∈[0,s−1] and j∈[0,M−1].(iv)Suppose that $A\in C^{N\times M}$ and $C\in C^{N\times M}$ are PBA and DQPBA, respectively. Then, $A^{T}\in C^{M\times N}$ and $C^{T}\in C^{M\times N}$ are the new PBA and DQPBA, respectively, where the superscript ${\left(\cdot \right)}^{T}$ denotes the transpose of the matrix.

Utilizing the methods above, we can construct the required PBA for the superimposed array P. We provide an example of generating a $P\in C^{8\times 8}$ that satisfies (Equation 18).

For PBA $A\in C^{4\times 4}$ and QPBA $B\in C^{4\times 4}$ given as
$$A=\left[\begin{array}{cccc} 1 & 1 & 1 & -1\\ 1 & 1 & 1 & -1\\ 1 & 1 & 1 & -1\\ -1 & -1 & -1 & 1 \end{array}\right],\;B=\left[\begin{array}{cccc} 1 & 1 & 1 & 1\\ 1 & 1 & -1 & -1\\ 1 & -1 & 1 & -1\\ 1 & 1 & -1 & -1 \end{array}\right]$$

Utilizing the iterative construction method (i), after one iteration, $P\in C^{8\times 8}$ can be obtained as
(30)$$P=\left[\begin{array}{cccccccc} 1 & 1 & 1 & 1 & 1 & 1 & -1 & 1\\ 1 & 1 & 1 & 1 & 1 & -1 & -1 & -1\\ 1 & 1 & 1 & -1 & 1 & 1 & -1 & -1\\ -1 & 1 & -1 & 1 & -1 & -1 & 1 & -1\\ 1 & -1 & 1 & -1 & 1 & -1 & -1 & -1\\ 1 & -1 & 1 & -1 & 1 & 1 & -1 & 1\\ 1 & -1 & 1 & 1 & 1 & -1 & -1 & 1\\ -1 & -1 & -1 & -1 & -1 & 1 & 1 & 1 \end{array}\right].$$

The autocorrelation of $P\in C^{8\times 8}$ given in (Equation 30) is calculated as $\mathrm{sum}\left(P\circ P_{k_{j},l_{j}}^{*}\right)$, where $k_{j},l_{j}\in \left[0,7\right]$. The autocorrelation result is shown in Figure 3. It can be seen that $\mathrm{sum}\left(P\circ P_{k_{j},l_{j}}^{*}\right)\neq 0$ if and only if $\left(k_{j},l_{j}\right)=\left(0,0\right)$. Therefore, the autocorrelation of P satisfies (Equation 18), which means it is a PBA that can be used in the proposed method.

For OTFS systems, the number of Doppler bins, *M*, is typically greater than the number of delay bins, *N*. Therefore, $T=\mathrm{PBA}\left(N,N\right)$ is selected in this paper. Moreover, to ensure the autocorrelation of the superimposed array P, the frame structure should satisfy $M=n_{\mathrm{PBA}}N$, where $n_{\mathrm{PBA}}$ is a positive integer. In summary, to satisfy the ideal autocorrelation condition in (Equation 18), the superimposed array P should be composed of $n_{\mathrm{PBA}}$ identical PBAs and can be expressed as
(31)$$P=[\underset{n_{\mathrm{PBA}}}{\underset{︸}{T\;T\;T\;\ldots \;T}}].$$

It is easy to prove that the superimposed array P in (Equation 31) is a PBA.

### 3.3. Rectangular Pulse Conditions

In practice, ideal pulses cannot be achieved [11]. For better compatibility with OFDM systems, $g_{\mathrm{tx}}$ and $g_{\mathrm{rx}}$ are usually set as rectangular pulses. The accuracy of SPBA-OTFS channel estimation is decreased by the additional interference $\alpha_{i}\left[k,l\right]$ introduced by the imperfect biorthogonality of the rectangular pulses according to (Equation 8).

We now investigate SPBA-OTFS channel estimation with rectangular pulses. Note that $\alpha_{i}\left[k,l\right]=e^{j2\pi lk_{i}/MN\;}$ when $l_{i}\leq l<M$. That is, only phase shifts exist under this condition. Additionally, the maximum delay is much less than the symbol period, meaning that $l_{m}\ll M$. Hence, for an SPBA-OTFS system with rectangular pulses, the correlation interval in (Equation 12) is reduced from 0≤l<M to $l_{m}\leq l<M$, and (Equation 12) can be rewritten as
(32)$$z\left[k_{J},l_{J}\right]=\frac{\mathrm{sum}\left(Y\circ \Gamma_{J}\cdot R\right)}{MN}\;,$$
where $R={\left[\begin{array}{cc} 0 & I \end{array}\right]}^{T}\in C^{M\times \left(M-l_{m}\right)}$, with I being the identity matrix.

Next, for $\alpha_{i}\left[k,l\right]=e^{j2\pi lk_{i}/MN\;}$, when $l_{i}\leq l<M$, we redefine the local PBA $\Gamma_{J}^{\mathrm{Rec}}$ associated with search unit *J* in the case of rectangular pulses as
(33)$$\Gamma_{J}^{\mathrm{Rec}}=\frac{e^{-j2\pi \frac{\left(l-l_{J}\right)k_{J}}{MN}}P_{k_{J},l_{J}}^{*}}{\beta \rho }.$$

Based on the analysis above, we need to modify only steps 4 and 5 in Algorithm 1 to adapt it to the rectangular pulse scenario. By adopting (Equation 33) to improve the local PBA generation in step 4 and reducing the correlation interval to $l_{m}\leq l<M$ on the delay axis in step 5, Algorithm 1 can be modified for use in the rectangular pulse scenario.

The relationship between the channel estimation MSE with rectangular pulses and that with ideal pulses is given by
(34)$$\mathrm{MSE}{\left(\hat{h}\right)}_{\mathrm{Rec}}=\frac{M}{M-l_{m}}\mathrm{MSE}\left(\hat{h}\right),$$
where $\mathrm{MSE}(\hat{h})$ is the MSE with ideal pulses in (Equation 22). The threshold satisfies $\gamma_{\mathrm{Rec}}^{2}\propto \mathrm{MSE}{\left(\hat{h}\right)}_{\mathrm{Rec}}$. Equation (Equation 34) shows that due to the effect of the imperfect biorthogonality of rectangular pulses, the accuracy of SPBA-OTFS channel estimation with rectangular pulses is slightly lower than that with ideal pulses.

### 3.4. Minimum Mean Square Error Estimator

Recall from (Equation 21) that the threshold method of estimation is influenced by the interference term $v_{q}$. To reduce this interference and achieve more accurate channel estimation, we consider the characteristics of wireless channels and use a minimum MSE (MMSE) estimator to smooth the threshold method estimation ${\hat{h}}_{q}$.

The MMSE channel estimate is given as
(35)$${\bar{h}}_{q}=\left(\frac{\sigma_{h_{q}}^{2}}{\sigma_{h_{q}}^{2}+\sigma_{v_{q}}^{2}}\right){\hat{h}}_{q},$$
where $\sigma_{v_{q}}^{2}$ is the MSE of ${\hat{h}}_{q}$ in the threshold method, as shown in (Equation 22). Similar to (Equation 21), the MMSE channel estimation result can be given as
(36)$${\bar{h}}_{q}=h_{q}+{\bar{v}}_{q},$$
where ${\bar{v}}_{q}$ denotes the channel estimation error with the MMSE estimator.

Because of the orthogonality of the MMSE estimator [17], the MSE of $h_{q}$ in (Equation 35) can be expressed as
(37)$$\sigma_{{\bar{v}}_{q}}^{2}=\frac{\sigma_{v_{q}}^{2}\sigma_{h_{q}}^{2}}{\sigma_{v_{q}}^{2}+\sigma_{h_{q}}^{2}},$$
while the MSE of MMSE channel estimation is
(38)$$\sigma_{\mathrm{MMSE}}^{2}=\sum_{q=1}^{P}\sigma_{{\bar{v}}_{q}}^{2}=\sum_{q=1}^{P}\frac{\sigma_{v_{q}}^{2}\sigma_{h_{q}}^{2}}{\sigma_{v_{q}}^{2}+\sigma_{h_{q}}^{2}}.$$

As $\sum_{q=1}^{P}\sigma_{h_{q}}^{2}=\sigma_{H}^{2}$, by utilizing the Lagrange multiplier method, we can obtain the following upper bound on (Equation 38):(39)$$\begin{array}{cc} \sigma_{\mathrm{MMSE}}^{2}\leq {\bar{\sigma^{2}}}_{\mathrm{MMSE}} & =\frac{\sigma_{H}^{2}\sigma_{v_{q}}^{2}}{\sigma_{v_{q}}^{2}+\sigma_{H}^{2}/P}\\ \; & =\frac{\sigma_{H}^{2}\cdot \mathrm{MSE}\left(\hat{h}\right)}{\mathrm{MSE}\left(\hat{h}\right)+\sigma_{H}^{2}/P}. \end{array}$$

This upper bound is obtained when the power of each path is equal, which means that $\sigma_{h_{1}}^{2}=\sigma_{h_{2}}^{2}=\cdots =\sigma_{h_{P}}^{2}$. By comparing (Equation 23) and (Equation 39), it can be seen that $\sigma_{\mathrm{Thr}}^{2}\geq {\bar{\sigma^{2}}}_{\mathrm{MMSE}}$. This result shows that the MMSE estimator can successfully improve the channel estimation accuracy.

## 4. Spectral Efficiency and Optimal Power Ratio

Recall from Section 2 that the superimposed symbol x[k,l] is formed by superimposing the PBA symbol $x_{p}\left[k,l\right]$ and data symbol $x_{d}\left[k,l\right]$. Under the condition that the total power is constant, the power ratio β∈[0,1] between PBA symbol and superimposed symbol will have a significant effect on the channel estimation performance and SE of the proposed method. Similar to [17], to minimize BER and maximize SE, we derive the optimal power ratio for the proposed SPBA-OTFS systems.

### 4.1. Spectrum Efficiency Analysis

The ultimate goal of designing superimposed PBA is to improve the SE. In OTFS systems, SE is given as [17]
(40)$$\mathrm{SE}=\left(1-\eta \right)\log_{2}\left(1+{\mathrm{SINR}}_{\mathrm{eff}}\right),$$
where η denotes the pilot overhead and ${\mathrm{SINR}}_{\mathrm{eff}}$ denotes the effective SINR for the OTFS system.

In the proposed method, because the PBA used as the pilot is superimposed on the transmitted symbol matrix, it holds that η=0 when compared to $\eta =\left(2l_{m}+1\right)\left(4k_{m}+1\right)/MN$ for the integer Doppler condition and $\eta =\left(2l_{m}+1\right)/M$ for the fractional Doppler condition in [13]. Thus, the superimposed PBA fundamentally improves the SE of an OTFS system.

Next, we derive an expression for the effective SINR in SPBA-OTFS systems using the received signal decoupled in the DD domain.

In the SPBA-OTFS system, the receiver decouples PBA and data by utilizing the estimated channel parameters in the DD domain. According to (Equation 9), the received symbol matrix $\bar{Y}$ after decoupling can be expressed as
(41)$$\begin{array}{cc} \bar{Y} & =Y-\sum_{q=1}^{P}{\bar{h}}_{q}e^{-j2\pi \frac{l_{q}k_{q}}{MN}}P_{k_{q},l_{q}}\\ \; & =\sum_{q=1}^{P}h_{q}e^{-j2\pi \frac{l_{q}k_{q}}{MN}}\left(D_{k_{q},l_{q}}+P_{k_{q},l_{q}}\right)\\ \; & +W-\sum_{q=1}^{P}{\bar{h}}_{q}e^{-j2\pi \frac{l_{q}k_{q}}{MN}}P_{k_{q},l_{q}}. \end{array}$$

Recall from (Equation 36) that ${\bar{h}}_{q}=h_{q}+{\bar{v}}_{q}$, (Equation 41) can be rewritten as
(42)$$\begin{array}{cc} \bar{Y} & =\sum_{q=1}^{P}\left({\bar{h}}_{q}-{\bar{v}}_{q}\right)e^{-j2\pi \frac{l_{k}k_{q}}{MN}}\left(D_{k_{q},l_{q}}+P_{k_{q},l_{q}}\right)\\ \; & +W-\sum_{q=1}^{P}{\bar{h}}_{q}e^{-j2\pi \frac{l_{l}k_{q}}{MN}}P_{k_{q},l_{q}}\\ \; & =\underset{A}{\underset{︸}{\sum_{q=1}^{P}{\bar{h}}_{q}e^{-j2\pi \frac{l_{q}k_{q}}{MN}}D_{k_{q},l_{q}}}}+\underset{B}{\underset{︸}{\sum_{q=1}^{P}\left(-{\bar{v}}_{q}\right)e^{-j2\pi \frac{l_{q}k_{q}}{MN}}D_{k_{q},l_{q}}}}\\ \; & +\underset{C}{\underset{︸}{\sum_{q=1}^{P}\left(-{\bar{v}}_{q}\right)e^{-j2\pi \frac{l_{q}k_{q}}{MN}}P_{k_{q},l_{q}}}}+W. \end{array}$$

We use matrices A, B and C to simplify (Equation 42), where A, B and C denote the effective signal, the interference introduced by data and the interference introduced by PBA, respectively. For the linear MMSE estimator, the estimation error is orthogonal to the observations [17]. Thus, the average effective SINR of each DD domain symbol is given by
(43)$${\mathrm{SINR}}_{\mathrm{eff}}=\frac{E\left\{AA^{*}\right\}}{E\left\{BB^{*}\right\}+E\left\{CC^{*}\right\}+E\left\{WW^{*}\right\}},$$
where
(44)$$E\left\{AA^{*}\right\}=MN\sum_{q=1}^{P}\sigma_{{\bar{h}}_{q}}^{2}\left(1-\beta \right)\rho ,$$
(45)$$E\left\{BB^{*}\right\}+E\left\{WW^{*}\right\}=MN\left(\sum_{q=1}^{P}\sigma_{{\bar{v}}_{q}}^{2}\left(1-\beta \right)\rho ,+\sigma_{\omega }^{2}\right).$$

For $E\{CC^{*}\}$, we use the Cauchy–Schwarz inequality to derive its upper bound as follows:(46)$$\begin{array}{cc} E\left\{CC^{*}\right\} & =E\left\{\sum_{q=1}^{P}{\bar{v}}_{q}e^{-j2\pi \frac{l_{q}k_{q}}{MN}}P_{k_{q},l_{q}}\sum_{q=1}^{P}{\bar{v}}_{q}^{*}e^{j2\pi \frac{l_{q}k_{q}}{MN}}P_{k_{q},l_{q}}^{*}\right\}\\ \; & \leq E\left\{\sum_{q=1}^{P}{\bar{v}}_{q}{\bar{v}}_{q}^{*}\right\}E\left\{\sum_{q=1}^{P}P_{k_{q},l_{q}}P_{k_{q},l_{q}}^{*}\right\}\\ \; & =MNP\beta \rho \sum_{q=1}^{P}\sigma_{{\bar{v}}_{q}}^{2}. \end{array}$$

By substituting (Equation 44), (Equation 45) and (Equation 46) into (Equation 43), the effective SINR is obtained as follows:(47)$$\begin{array}{cc} {\mathrm{SINR}}_{\mathrm{eff}} & \geq \frac{\sum_{q=1}^{P}\sigma_{{\bar{h}}_{q}}^{2}\left(1-\beta \right)\rho }{\sum_{q=1}^{P}\sigma_{{\bar{v}}_{q}}^{2}\left(1-\beta \right)\rho ,+\sigma_{\omega }^{2}+P\beta \rho \sum_{q=1}^{P}\sigma_{{\bar{v}}_{q}}^{2}}\\ \; & =\frac{\left(\sigma_{H}^{2}-\sigma_{\mathrm{MMSE}}^{2}\right)\left(1-\beta \right)\rho }{\sigma_{\mathrm{MMSE}}^{2}\left(1-\beta \right)\rho +P\sigma_{\mathrm{MMSE}}^{2}\beta \rho +\sigma_{\omega }^{2}}. \end{array}$$

### 4.2. Optimal Power Ratio

According to (Equation 40), SE is proportional to ${\mathrm{SINR}}_{\mathrm{eff}}$. To maximize the SE, we now maximize the ${\mathrm{SINR}}_{\mathrm{eff}}$ in (Equation 47). The upper bound on the MMSE from (Equation 39) is substituted into (Equation 47) to obtain the lower bound on the effective SINR, as follows:(48)$${\underline{\mathrm{SINR}}}_{\mathrm{eff}}=\frac{\left(\sigma_{H}^{2}-{\bar{\sigma^{2}}}_{\mathrm{MMSE}}\right)\left(1-\beta \right)\gamma_{\mathrm{SNR}}}{{\bar{\sigma^{2}}}_{\mathrm{MMSE}}\left(1-\beta \right)\gamma_{\mathrm{SNR}}+P{\bar{\sigma^{2}}}_{\mathrm{MMSE}}\beta \gamma_{\mathrm{SNR}}+1},$$
where $\gamma_{\mathrm{SNR}}=\rho /\sigma_{\omega }^{2}$. According to (Equation 48), the lower bound on the effective SINR is the function of the power ratio β. To maximize the ${\mathrm{SINR}}_{\mathrm{eff}}$ and calculate the optimal power ratio, we take the derivative of ${\underline{\mathrm{SINR}}}_{\mathrm{eff}}$ and equate it to zero. Under this condition, the lower bound on the effective SINR will reach the optimal value. Using (Equation 39), the derivative result is as follows:(49)$$\frac{\partial {\underline{\mathrm{SINR}}}_{\mathrm{eff}}}{\partial \beta }=\frac{A\beta^{2}-B\beta +C}{{\left(f(\beta )\right)}^{2}}=0,$$
where
(50)$$\left\{\begin{array}{c} A=MN\left(-\sigma_{H}^{6}\gamma_{\mathrm{SNR}}^{3}MN+2\sigma_{H}^{6}\gamma_{\mathrm{SNR}}^{3}P-\sigma_{H}^{6}\gamma_{\mathrm{SNR}}^{3}P^{2}\right.\\ \left.\;+\sigma_{H}^{8}\gamma_{\mathrm{SNR}}^{4}P\right)\\ B=MN\left(2\sigma_{H}^{4}\gamma_{\mathrm{SNR}}^{2}P+4\sigma_{H}^{6}\gamma_{\mathrm{SNR}}^{3}P+2\sigma_{H}^{8}\gamma_{\mathrm{SNR}}^{4}P\right)\\ C=MN\left(\sigma_{H}^{4}\gamma_{\mathrm{SNR}}^{2}P+2\sigma_{H}^{6}\gamma_{\mathrm{SNR}}^{3}P+\sigma_{H}^{8}\gamma_{\mathrm{SNR}}^{4}P\right)\\ \begin{array}{cc} f(\beta ) & =\beta \sigma_{H}^{2}MN\gamma_{\mathrm{SNR}}\\ \; & -\beta \sigma_{H}^{2}P^{2}\gamma_{\mathrm{SNR}}\left(\left(\beta -1\right)\sigma_{H}^{2}\gamma_{\mathrm{SNR}}-1\right)\\ \; & +P{\left(\left(\beta -1\right)\sigma_{H}^{2}\gamma_{\mathrm{SNR}}-1\right)}^{2} \end{array} \end{array}.\right.$$

According to (Equation 50), because of f(β)≠0, the positive solution to (Equation 49) is the optimal power ratio, given as
(51)$$\beta_{\mathrm{opt}}=\frac{B-\sqrt{B^{2}-4AC}}{2A}.$$

When the optimal power ratio above is used, the proposed method can achieve better channel estimation and SE performance.

## 5. Simulation Results

In this section, we first demonstrate the rationality of adopting the optimal power ratio $\beta_{\mathrm{opt}}$. Utilizing $\beta_{\mathrm{opt}}$, we evaluate the performance of the proposed SPBA-OTFS channel estimation method in terms of SE, channel estimation and PAPR performance. The OTFS frame size in the DD domain is set to N=128 and M=512. The symbols are drawn from the QPSK constellation. The carrier frequency is 4.9 GHz, and the subcarrier spacing is 30 kHz. The channel model is the Extended Typical Urban (ETU) model [22], where the number of propagation paths is P=9. For each path, its Doppler shift is generated in accordance with the Jakes formula [13]. The maximum delay index is $l_{m}=77$, and the maximum Doppler index is $k_{m}=10$, corresponding to a UE speed of 500 km/h. For the proposed SPBA method, we set the power of each superimposed symbol to ρ=1 and the threshold to $\gamma^{2}=12.25\mathrm{MSE}$.

### 5.1. Optimal Power Ratio and SE Performance

Figure 4 shows the optimal power ratio $\beta_{\mathrm{opt}}$ under different frame structures. In this subsection, the channel power levels satisfy the normalization condition $\sum_{i=1}^{P}\sigma_{h_{i}}^{2}=1$. The results show that $\beta_{\mathrm{opt}}$ is proportional to SNR. This is because under low-SNR conditions, the main interference term in the proposed method is the noise term, and the system needs to reduce the power allocated to superimposed symbols to improve the noise resistance of data symbols; however, under high-SNR conditions, to ensure the channel estimation ability, the system should allocate more power to superimposed symbols. Figure 4 also shows that $\beta_{\mathrm{opt}}$ is inversely proportional to the frame structure size. The main reason is that as the frame structure size increases, the structure size of the superimposed PBAs also increases, and when performing the correlation operation (Equation 12), less energy needs to be allocated to PBAs to produce significant correlation peaks. Thus, $\beta_{\mathrm{opt}}$ can be reduced accordingly.

Figure 5 shows the effects of $\beta_{\mathrm{opt}}$ on SINR and BER with ideal MMSE estimation. It can be seen from Figure 5a that around $\beta_{\mathrm{opt}}$, the SINR will achieve its maximum value, while the BER will simultaneously achieve its minimum value as shown in Figure 5b. This verifies the rationality of adapting the optimal power ratio.

In Figure 6, we compare the SE performance of the proposed SPBA method with that of the EP method from [13]. As a reference, we also show the ideal SE performance, corresponding to the case in which channel information is already known by the system, meaning that no pilot is needed. It can be seen that the proposed SPBA method can achieve nearly ideal SE performance even at the lower bound. Compared with the EP design, because the pilot is superimposed onto the data matrix rather than embedded in it along with the protected symbols, the proposed method can achieve significantly improved SE performance. This improvement reaches 23.84% when SNR = 15 dB, when compared to the upper bound of EP design.

### 5.2. Channel Estimation Performance

The channel estimation performance can mainly be evaluated in terms of channel estimation normalized MSE (NMSE) and BER performance.

In Figure 7, we compare the NMSE performance of SPBA and EP methods. The NMSE is defined as
(52)$$\mathrm{NMSE}=E\left({∥\bar{h}-h∥}_{2}^{2}\right)/E\left({∥h∥}_{2}^{2}\right),$$
where $\bar{h}$ is the vector form of the MMSE channel estimation result ${\bar{h}}_{q}$. Additionally, for the SPBA method, the theoretical NMSE is given for reference. We set the power ratio to $\beta_{\mathrm{opt}}$ for the SPBA method and the pilot SNR to ${\mathrm{SNP}}_{p}=40\;\mathrm{dB}$ for the EP method. From Figure 7, it can be seen that the simulated NMSE curve *SPBA(Simu)* is basically consistent with the theoretical NMSE curve *SPBA(Theo)*, which confirms our analyses in Section 3. Regarding the comparison between the curves *EP(Simu)* and *SPBA(Simu)*, as the SNR increases, the NMSE performance of the proposed SPBA method approaches that of the EP method. The reason why the NMSE performance of the SPBA method is slightly inferior to that of the EP method can be explained as follows. As the SNR increases, the noise interference $v_{q,\omega }$ in the SPBA method is reduced, and the main interference term becomes $v_{q,d}$, which is the interference between superimposed PBA and transmitted symbols. In the EP method, however, the protected symbols eliminate this interference, at a high cost in terms of SE.

In Figure 8, we show the BER performance based on the MP detector in [11]. We assume that each transmitted OTFS symbol has a power ratio of $\beta_{\mathrm{opt}}$ for the SPBA method and ${\mathrm{SNP}}_{p}=40\;\mathrm{dB}$ for the EP method. Furthermore, the BER performance of the SPBA method is given with the threshold estimator in Section 3.1 and with the ideal MMSE estimator in Section 3.4, corresponding to the curves labeled *SPBA(Thr)* and *SPBA(MMSE)*, respectively. It can be seen in Figure 8 that with prior channel information, the MMSE estimator shows better performance than the threshold estimator. Compared with that of the EP method, due to the interference term $v_{q,d}$ mentioned above, the BER performance of the SPBA method is close but slightly inferior, with a loss of approximately 0.75 dB for the threshold estimator and 0.2 dB for the ideal MMSE estimator. However, as mentioned before, the proposed SPBA method has a significantly higher SE than the EP method.

## 6. Conclusions

In this paper, we propose a superimposed PBA-aided channel estimation method with high SE for OTFS systems. In the proposed method, a PBA is superimposed onto data symbols as the pilot in the DD domain. By exploiting the perfect autocorrelation of PBA, channel estimation is performed through linear search and a threshold-based method. The delay-Doppler indices and complex gains of channels are estimated in the same frame, so the SPBA method has better delay-Doppler varying adaptability. Additionally, the optimal power of the superimposed PBA is derived to maximize SE. The results of analyses and simulations show that despite a slight BER degradation, the proposed SPBA-OTFS method significantly improves the SE.

## Figures and Tables

**Figure 1 entropy-25-01163-f001:**
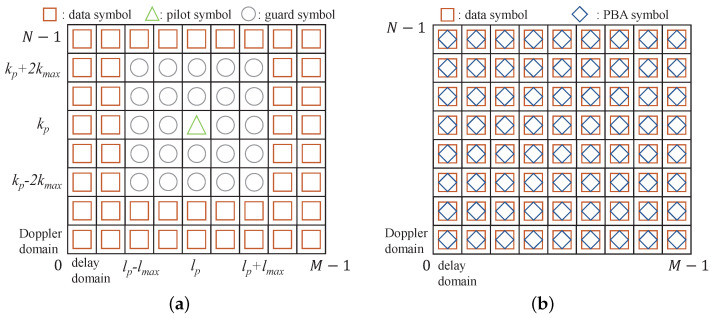
The frame structure of (**a**) EP-OTFS systems and (**b**) SPBA-OTFS systems.

**Figure 2 entropy-25-01163-f002:**
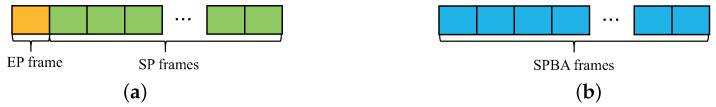
The transmitted data frames of (**a**) SP-OTFS systems and (**b**) SPBA-OTFS systems.

**Figure 3 entropy-25-01163-f003:**
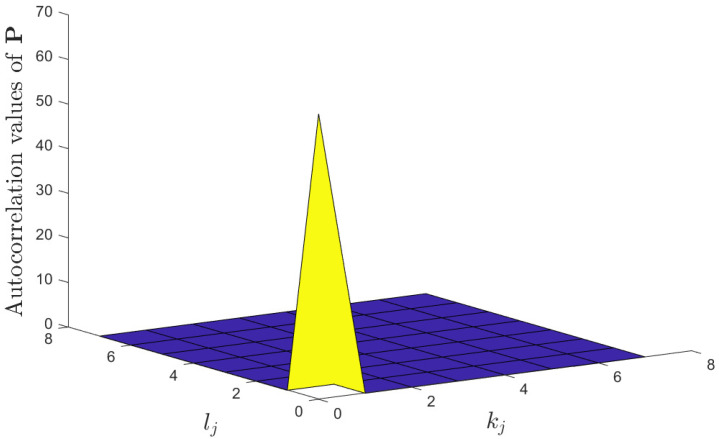
Autocorrelation value of P given in (Equation 30).

**Figure 4 entropy-25-01163-f004:**
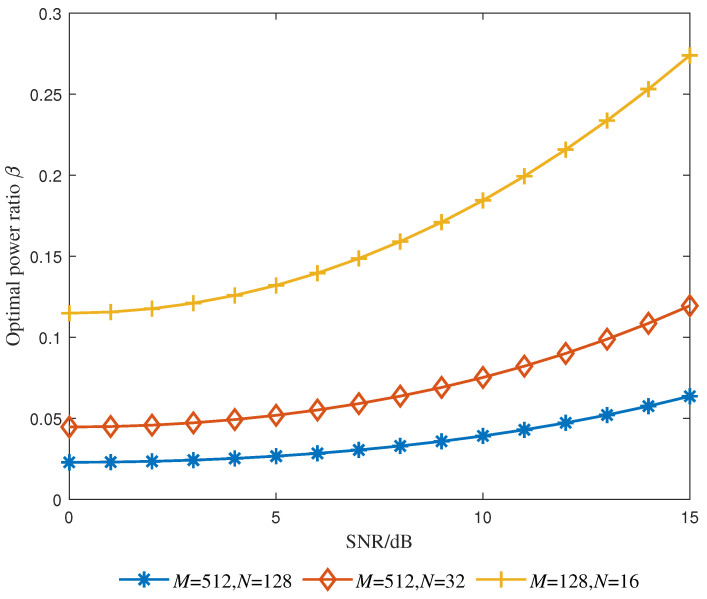
Optimal power ratio versus SNR.

**Figure 5 entropy-25-01163-f005:**
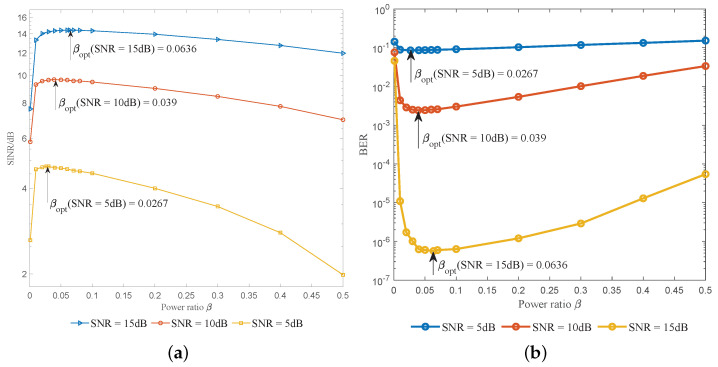
Effects of the optimal power ratio. (**a**) The effect of the optimal power ratio on the SINR. (**b**) The effect of the optimal power ratio on the BER with SNR = 10 dB.

**Figure 6 entropy-25-01163-f006:**
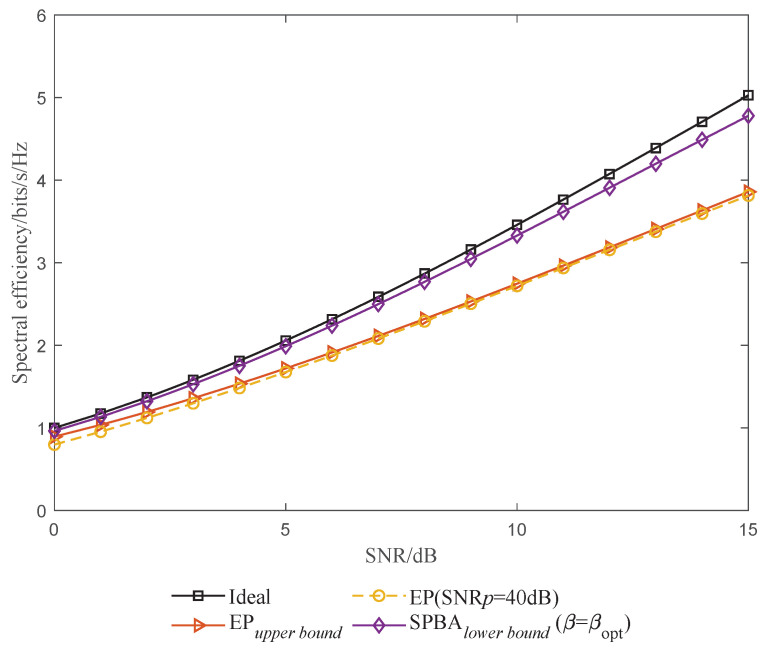
SE performance versus SNR.

**Figure 7 entropy-25-01163-f007:**
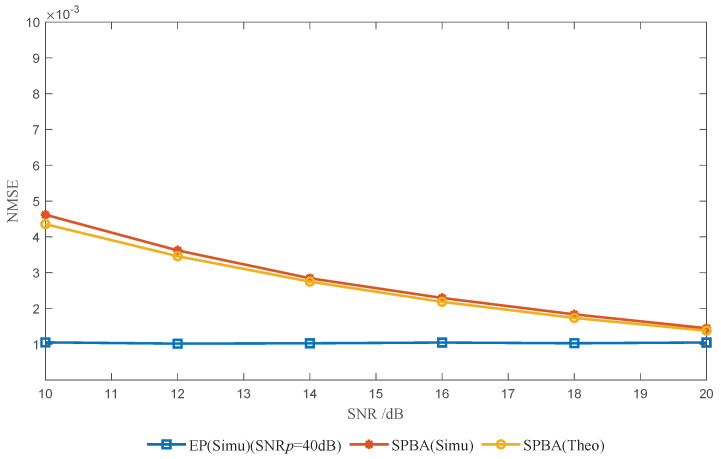
Channel estimation NMSE versus SNR.

**Figure 8 entropy-25-01163-f008:**
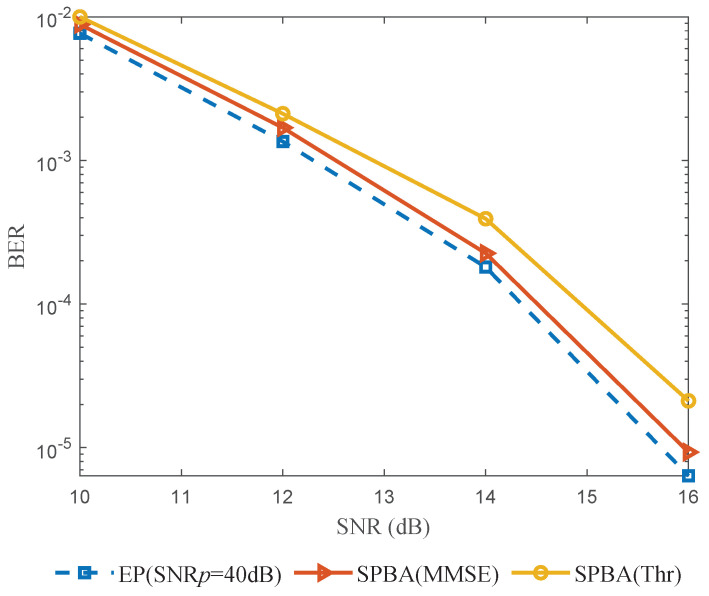
BER performance versus SNR.

## Data Availability

Not applicable.

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
