# Peer review of "Superimposed Perfect Binary Array-Aided Channel Estimation for OTFS Systems"

_entropy, 2023, doi:10.3390/e25081163_

Round 1

Reviewer 1 Report

Please find my comments attached as PDF file.

Author Response

Please find our reply attached as PDF file.

Reviewer 2 Report

The authors propose a high SE channel estimation method that superimposes the perfect binary array (PBA) on data symbols as the pilot. Utilizing the perfect autocorrelation of PBA, channel estimation is performed based on a linear search to find the correlation peaks, which include both the delay-Doppler taps information and the complex channel gain in the same superimposed PBA frame.

2. Please elaborate the technical features and technique efficacy of SE channel estimation method in detail.

3. Please compare the contributions of the proposed technology to related technologies, in detail. Consider adding a detailed discussion concerning related technologies.

4. In the figure 1, the frame structure of (a) EP-OTFS systems from [13] and (b) SPBA-OTFS systems, should be elaborated in detail.

5.In the figure 2, the transmitted data frames of (a) SP-OTFS systems from [17] and (b) SPBA-OTFS systems, should be elaborated in detail.

6. In the figure3,autocorrelation value sum _P â—¦ Pkj ,lj , where kj, lj [0, 7], should be elaborated in detail.

7.Please thoroughly revise the language before your submission.

 Extensive editing of English language required.

Author Response

(The authors gave the same response as above.)
